# The Impacts of *Lactiplantibacillus plantarum* on the Functional Properties of Fermented Foods: A Review of Current Knowledge

**DOI:** 10.3390/microorganisms10040826

**Published:** 2022-04-15

**Authors:** Birsen Yilmaz, Sneh Punia Bangar, Noemi Echegaray, Shweta Suri, Igor Tomasevic, Jose Manuel Lorenzo, Ebru Melekoglu, João Miguel Rocha, Fatih Ozogul

**Affiliations:** 1Department of Nutrition and Dietetics, Cukurova University, Sarıcam, 01330 Adana, Turkey; ebrumelekoglu@hotmail.com; 2Department of Food, Nutrition and Packaging Sciences, Clemson University, Clemson, SC 29631, USA; snehpunia69@gmail.com; 3Centro Tecnológico de la Carne de Galicia, Adva. Galicia no. 4, Parque Tecnológico de Galicia, San Cibrao das Viñas, 32900 Ourense, Spain; noemiechegaray@ceteca.net (N.E.); jmlorenzo@ceteca.net (J.M.L.); 4Department of Food Engineering, National Institute of Food Technology Entrepreneurship and Management, Sonipat 131028, India; shwetasuri94@gmail.com; 5Faculty of Agriculture, University of Belgrade, Nemanjina 6, 11080 Belgrade, Serbia; tbigor@agrif.bg.ac.rs; 6Área de Tecnología de los Alimentos, Facultad de Ciencias de Ourense, Universidad de Vigo, 32004 Ourense, Spain; 7LEPABE–Laboratory for Process Engineering, Environment, Biotechnology and Energy, Faculty of Engineering, University of Porto, Rua Dr. Roberto Frias, 4200-465 Porto, Portugal; jmfrocha@fe.up.pt; 8ALiCE–Associate Laboratory in Chemical Engineering, Faculty of Engineering, University of Porto, Rua Dr. Roberto Frias, 4200-465 Porto, Portugal; 9Department of Seafood Processing Technology, Faculty of Fisheries, Cukurova University, Balcali, 01330 Adana, Turkey; fozogul@cu.edu.tr

**Keywords:** *Lactiplantibacillus plantarum*, lactic acid bacteria, fermented food, food industry, safety aspects

## Abstract

One of the most varied species of lactic acid bacteria is *Lactiplantibacillus plantarum* (*Lb. plantarum*), formerly known as *Lactobacillus plantarum*. It is one of the most common species of bacteria found in foods, probiotics, dairy products, and beverages. Studies related to genomic mapping and gene locations of *Lb. plantarum* have shown the novel findings of its new strains along with their non-pathogenic or non-antibiotic resistance genes. Safe strains obtained with new technologies are a pioneer in the development of new probiotics and starter cultures for the food industry. However, the safety of *Lb. plantarum* strains and their bacteriocins should also be confirmed with in vivo studies before being employed as food additives. Many of the *Lb. plantarum* strains and their bacteriocins are generally safe in terms of antibiotic resistance genes. Thus, they provide a great opportunity for improving the nutritional composition, shelf life, antioxidant activity, flavour properties and antimicrobial activities in the food industry. Moreover, since some *Lb. plantarum* strains have the ability to reduce undesirable compounds such as aflatoxins, they have potential use in maintaining food safety and preventing food spoilage. This review emphasizes the impacts of *Lb. plantarum* strains on fermented foods, along with novel approaches to their genomic mapping and safety aspects.

## 1. Introduction

*Lactiplantibacillus plantarum* (formerly *Lactobacillus plantarum*) is one of the Gram-positive lactic acid bacteria (LAB) species [1]. *Lb. plantarum* has high ecological and metabolic adaptability that exists widely in a range of habitats including fermented dairy products, sourdoughs, fruits, vegetables, cereals, meat, fish, and the mammalian gastrointestinal tract [2]. In the production of various fermented foods, *Lb. plantarum* has been widely used as a starter culture that improves the flavor, texture and organoleptic properties of food products [3]. It also provides the functional properties of the fermented foods by producing a variety of bioactive components, including exopolysaccharides, γ-aminobutyric acid, riboflavin, folic acid, and vitamin B_12_ [4,5,6]. Moreover, *Lb. plantarum* is one of the most used bacterial strains in food processing and preservation as a food preservative through the production of diverse and potent bacteriocins (class I and II) and organic acid [7,8]. In particular, bacteriocins have a broad antimicrobial activity spectrum against Gram-positive and Gram-negative bacteria [9]. *Lb. plantarum* has a qualified presumption of safety (QPS) from the European Food Safety Authorities (EFSA) and is “generally recognized as safe” (GRAS) status by the United States Food and Drug Administration (US FDA) [10]. Since most of the LAB species are known as GRAS and QPS, bacteriocins are expected to be safe to use in the food industry as bio-preservatives [11,12].

It has greatly hastened the discovery of new strains of interest in the food industry and biotechnology since probiotic phenotypes may be traced back to specific genes and genetic clusters [13]. The whole-genome sequencing tries to explain genomic mapping of *Lactobacillus* species, isolated from different fermented foods. The characterisation of bacteriocin and the identification of probiotic genes can be explained through the studies. According to the genome sequence analysis, no pathogenic or antibiotic resistance genes were identified in *Lb. plantarum*. However, it has been reported that the *Lb. plantarum* genome (varies from 3.0 to 3.3 Mb) is greater than the other LAB species [14,15].

This paper focuses on the genotypic characterization, functional properties, and safety aspects of *Lb. plantarum* and new research on foodomics of some functional fermented foods using *Lb. plantarum* in a broad perspective.

## 2. Genomic Mapping and Gene Locations of *Lactiplantibacillus plantarum*


*Lb. plantarum* is one of the promising LAB species, which is extensively utilized in the food industry for its use as a probiotic and starter culture [16]. Owing to its vast history of safe application in human foods, most LAB species, especially *Lb. plantarum,* are incorporated in the QPS recommendations of the European Food Safety Authority [17,18].

As per the literature, the *Lb. plantarum* genome (3.3 Mb) is greater than the distinctive genome of other LAB species (2–2.7 Mb). The larger genome size of *Lb. plantarum* advocates a very high level of genetic diversity within the species, which is attributed to this species’ nomadic life, inhabiting a wide variety of habitats and exhibiting great metabolic diversity [19,20,21]. Due to the high intraspecies diversity, it is difficult to classify the strains of *Lb. plantarum* based on simple characteristic traits. Previous studies of comparative genomic analysis have repetitively confirmed the progression of *Lb. plantarum* is not associated with the source of isolation or the geographic location of the strains belonging to this species [19]. Nonetheless, alterations in some gene clusters were found among *Lb. plantarum* strains. A comparison of 23 strains of *Lb. plantarum* showed that they evolved to comprise interspaced short palindromic repeats, antimicrobial action, and detoxification activity [22]. Six strains of *Lb. plantarum* were studied, and a significant difference was found in prophages, transposase, IS elements, and plantaricin biosynthesis genes among the strains. Furthermore, a high variation was observed in capsular plus extracellular polysaccharide biosynthesis genes [23].

A more recent study described the genomic properties of the *Lb. plantarum* strain UTNGt2 was obtained from wild copoazu (*Theobroma grandiflorum*), also known as white cacao. They also studied the variation in the genes of *Lb. plantarum* UTNGt2 strain through diverse hypervariable CRISPR (clustered regularly interspaced short palindromic repeats)/Cas systems. Based on the results of gene prediction and annotation, 9.4% of proteins were observed to be involved in carbohydrate transport as well as metabolism, 8.46% were involved in transcription, 2.36% were involved in defence mechanisms, and 0.5% carried out secondary metabolite biosynthesis, transport, and catabolism, whereas the remaining 25.11% had an unknown action. The genome study reveals the occurrence of genes engaged in riboflavin and folic acid production. Besides, the presence of CRISPR/Cas genes, phage sequences, the nonexistence of acquired antibiotic resistance genes, pathogenicity, and virulence factors indicated that the UTNGt2 is a safe strain. Its high antibacterial activity is associated with the existence of two bacteriocin clusters (class IIc), the sactipeptide class (contig 4) and the plantaricin E class (contig 22). The study demonstrates that UTNGt2 is a non-pathogenic, nonvirulent strain and can be used as a probiotic in food applications [24]. Similarly, the characterization of *Lb. plantarum* R23 and its bacteriocin were conducted. The genome sequence of *Lb. plantarum* was done by whole-genome sequencing (WGS). No pathogenic or antibiotic resistance genes were identified in *Lb. plantarum*. Four proteins that are 100% identical to Class II bacteriocins (Plantaricin E, Plantaricin F, Pediocin PA1 (Pediocin AcH), and Coagulin A) were detected through WGS analysis. The small (<6.5 kDa) R23 bacteriocin was observed to be stable at varying pH values (range 2–8), temperature (4–100 °C), detergents (all excluding Triton X100 as well as Triton X114 at 0.01 g/mL), and enzymes (catalase and α-amylase). In addition, they do not adsorb to producer cells, have a bacteriostatic mode of action, and their maximum activity (12,800 AU/mL) against the two *Listeria monocytogenes* strains is between 15–21 h of *Lb. plantarum* R23 growth. This study indicated that *Lb. plantarum* R23 is safe and promising as a bio-conservative culture because it produces stable bacteriocins [25]. 

Likewise, a group of researchers drafted the genomic sequence of *Lb. plantarum* L125. The entire genome of *Lb. plantarum* L125 comprises 3,354,135 bp, has a GC content of 44.34%, contains prophage regions, and does not contain CRISPR arrays. The 3220 predicted genes comprised protein-coding sequences (3024), pseudogenes (126), tRNA genes (62), rRNA genes (4), and ncRNAs (4). *Lb. plantarum* L125, usually isolated from meat-based foodstuffs, adapts to different niches, as indicated by the fact that 88 of its genes are mapped to the KEGG microbial metabolism in various environmental pathways. *Lb. plantarum* strains can colonize various habitats, including the human gastrointestinal tract, vegetables, meat, fish, dairy products, and other fermented foodstuffs (Figure 1). This kind of nomadic life of *Lb. plantarum* is reflected in the vast genetic diversity of the *Lb. plantarum* strain [13]. 

In a recent study, *Lb. plantarum* X7021 was isolated from the Chinese fermented stinky tofu. To examine the applicability of this strain in the food industry, researchers investigated genomic and metabolic properties using comparative genomics as well as transcriptional assays. The results show that *Lb. plantarum* X7021 is safe for application in food. *Lb. plantarum* X7021 was found to have 25 complete transporters of the phosphotransferase system and a strong proteolytic system so that it is adaptable to different foods [26]. In another study, the genomic changes in the probiotic *Lb. plantarum* P8 was studied in humans and rats. Experiments with the oral ingestion of P8 were carried out. During the experiment, the dynamics of P8 frequency in feces was monitored by qPCR. The amount of P8 in the feces was high during the period of use and decreased when the use was stopped. However, after a few days in both human and rat experiments, a slight increase or stable level of P8 in the fecal sample was observed, indicating that P8 may be temporarily widespread in the human and rat gastrointestinal tract [27]. A large-scale comparative genomic study of 455 *Lb. plantarum* genomes were conducted. Animal and dairy isolates showed significant deviations in phylogenetic distribution. The study revealed that dairy as well animal isolates have a number of environment-specific genes [28].

## 3. Gene Sequencing for *Lactiplantibacillus plantarum*

Around 560 *Lb. plantarum* genomes are available in the NCBI repository, 135 of which have been completed [29]. As per the past studies, the genome of *Lb. plantarum* strains is one of the largest genomes within the *Lactobacillus* group, with a GC content of approximately 44%. In addition, the number of coding sequences (CDS) is in the range of 1964 to 3526 for *Lb. plantarum* WHE92 and *Lb. plantarum* SRCM101258, respectively [30]. The foremost *Lb. plantarum* strain (WCFS1) was fully sequenced in 2003 and isolated from the saliva of human beings [31]. Extensive genome sequencing of the WCFS1 strain has provided the research fraternity with a deeper knowledge of this *Lb. plantarum* species. It has been the standard for additional in-silico research based on its gene prediction/annotation as a primary approach in predicting the phenotype [30]. *Lb. plantarum* is commonly observed in Indian fermented foods, for example, idli, dosa, and fermented sorghum-based products [32,33,34]. Nevertheless, it was not until 2009 that the strains obtained from fermented foodstuffs were sequenced [30]. 

The *Lb. plantarum* strain of food origin encodes genes for several stress-related proteins. The presence of the OpuC (osmoregulatory system), the chaperones groESgroEL and the hcrAdnaKdnaJGrpE operon, NADH oxidase, and peroxidase or thiol and manganese transporters confers an advantage on strains that allow them to survive under extreme gastrointestinal conditions [21,35]. In the context of the presence of the CRISPR-Cas system, the maximum *Lb. plantarum* stain shows the magnificence of the CRISPR-Cas system (Type II) with four genes, i.e., cas9, cas1, cas2, and csn2 [36].

*Lb. plantarum* has a lifestyle adaptation zone or lifestyle island in its genome. Areas are specific to *Lb. plantarum* mainly consists of sugar transport and utilization and performs extracellular functions that encode genes. This region seems to play a key role in the effective adaptation of *Lb. plantarum* to the environment [21]. The ability to ferment multiple sugars is one of the major properties of *Lb. plantarum* strains that have received special consideration. Their effective transport systems lead to high adaptability and the ability to live in diverse ecological conditions. The comparative study of the genome of *Lb. plantarum* isolates from different sources showed that most of the genes encoded in the “lifestyle adaptation zone” were not preserved among strains and encode genes predictive of plantaricin and exopolysaccharide biosynthesis. These results confirm the excellent plasticity of the *Lb. plantarum* genome, coupled with an effective metabolism, makes it a nomadic as well as a versatile species [30].

A group of researchers isolated *Lb. plantarum* from different sources and studied its genome sequencing. Recently, the genomic description of *Lb. plantarum* obtained from dahi and kinema showed the production of putative bacteriocin and probiotics [14]. In addition, *Lb. plantarum* Lp91 isolated from the human intestine [37] and JDARSH isolated from sheep milk were also sequenced for studying the genome [38]. Recently, *Lb. plantarum* ST was isolated from De’ang pickled tea. The strain ST genome was fully sequenced and examined through the PacBio RS II sequencing arrangement. *Lb. plantarum* ST is a potent probiotic strain and is highly tolerated in the simulated artificial gastrointestinal tract. It also exhibited robust antibacterial activity in antagonism tests. Hence, it can be used as a livestock probiotic. The *Lb. plantarum* ST genome consisted of one circular chromosome and seven plasmids. The complete genome is 3,320,817 bp, the size of the ring chromosome is 3,058,984 bp, guanine + cytosine (G±C) content is 44.76%, and contains 2945 protein-coding sequences (CDS) [39].

## 4. Evolutionary Patterns of *Lactiplantibacillus plantarum*

The *Lactobacillus* genus comprises more than 200 species known by phylogenetic and metabolic diversity that surpasses the usual bacterial family [40]. Current phylogenetic analysis based on the robust phylogenetic system of the genome core suggests that lactobacilli can be segmented into at least 24 phylogenetic groups [41]. The accessibility of *Lactobacillus* genome sequences provided a robust framework for large-scale phylogenetic and relative genomic analysis that could explain their evolution. Besides, population genomics and genetic analysis have enabled a comprehensive renewal of the evolutionary patterns of specific *Lactobacillus* species [40,41,42]. Literature indicates that monophyletic populations in *Lactobacilli* are due to adaptive evolution in diverse habitats, leading to the emergence of distinct lifestyles and a high degree of conservation of these species. The *Lb. plantarum* leads a free-living to nomadic life and it is stably found in various niches. The usual habitat of *Lb. plantarum* is fruit flies, the digestive tract of vertebrates, plants, as well as dairy items [43,44].

Like free-living lactobacilli, the large genomes of *Lb. plantarum* resembles improved metabolic flexibility. Moreover, strains of *Lb. plantarum* maintained conditional respiration capacity [45,46]. *Lb. plantarum* WCFS1 also promotes flexibility in diverse habitats by encoding a broad range of sugar uptake and utilization cassettes that allow organisms to grow on various carbon sources (e.g., plant-based- oligosaccharides and polysaccharides) [23]. Comparative genomic analysis of 54 strains of *Lb. plantarum* showed a lack of ecological specialization, which has already been suggested in earlier research. Strains of *Lb. plantarum* do not show distinct clustering according to origin. Studies clearly explain that genes involved in exopolysaccharide biosynthesis and sugar metabolism show the greatest variability among *Lb. plantarum* strains; however, there is no relationship [20,21].

*Lb. plantarum* is found in humans and animals, although it does not form a stable population in animal hosts. Yet it is a human-and animal-related niche with adaptive traits that contribute to sustainability. Additionally, some *Lb. plantarum* strains are highly resistant to gastric fluid and bile acids [42,47]. As *Lb. plantarum* originates in different habitats, it evolves in different ways, resulting in high intraspecific genetic diversity in this species. Strain diversity may benefit industrial applications, but it is disadvantageous in food safety. Comparative gene analysis is underway to investigate this more thoroughly [19,48]. Several studies have used various phenotypic and genotyping approaches such as amplified fragment length polymorphism (AFLP), random amplified polymorphic DNA (RAPD), polylocus sequence typing (MLST), and microarray-based comparative genomic hybridization of *Lb. plantarum* strains that showed genetic diversity. According to these studies, several strains of *Lb. plantarum* typically shows high conservatism of genes conducting protein and lipid synthesis or degradation and high diversity of genes carrying out sugar transport as well as catabolism [49,50,51,52,53].

## 5. The Impacts of *Lactiplantibacillus plantarum* on the Functional Properties of Fermented Foods

LAB are gram-positive bacteria that are common in nature and have a significant place in the food industry [54]. Lactic acid fermentation affects the taste and nutritional composition of foods (vitamins and amino acids) positively through producing organic acids, bacteriocins and volatile compounds, as well as helping to improve the organoleptic and qualitative characteristics (shelf life, food preservation and food safety) of foods [55,56,57]. *Lb. plantarum* has been reported to be present in the human gastrointestinal, vaginal and urogenital tracts. It also plays a role in the fermentation of many foods such as dairy products, vegetables, meat and wine [7,58].

The potential functional impacts of *Lb. plantarum* in the food industry are summarized in Table 1. For many years, *Lb. plantarum* has been widely used in food fermentation due to its non-harmful nature and improvement in the characteristics of fermented products [54,56]. In addition, some strains of *Lb. plantarum* have the ability to produce bacteriocins, which are particularly prominent with their antimicrobial properties and have food preservative applications [59]. Moreover, various *Lb. plantarum* strains have been shown to produce different antimicrobial compounds such as organic acids, hydrogen peroxide, and diacetyl [59]. Li et al., (2012) examined the antioxidant activity of *Lb. plantarum* strains isolated from traditional Chinese fermented foods and they reported that *Lb. plantarum* C88 (10^10^ CFU/mL) isolated from tofu can be used as a potential antioxidant in functional foods [60]. In another recent study, similarly, *Lb. plantarum* C88 isolated from tofu has been shown to reduce aflatoxin B1 toxicity [61]. Three different strains of *Lb. plantarum* (LP1, LP2 and LP3) showed high antibacterial activity against *Escherichia coli* ATCC 25922 and *Staphylococcus aureus* ATCC 25923 [62]. Furthermore, the antimicrobial effects of *Lb. plantarum* strains against food-borne pathogenic microorganisms were reported. Thus, *Lb. plantarum* 105 was found to have the strongest effect against *L. monocytogenes*, while *Lb. plantarum* 106 and 107 were found to have the strongest effect against *E. coli* O157:H7 [59]. These findings suggest that the use of *Lb. plantarum* in the food industry as a potential bio-control method against pathogenic microorganisms should be emphasized. *Lb. plantarum* is not only a more sustainable option (it can be used instead of artificial antimicrobial agents) but also has promising potential in the development of functional foods.

In addition to its antimicrobial effects, *Lb. plantarum* is known to improve flavour properties, preservation and/or enhancement of the product’s nutritional composition and health benefits, and extend shelf life. *Lactobacillus delbrueckii* subsp. *bulgaricus* and *Streptococcus thermophilus* are key factors in the final quality of fermented milk, especially in its aroma. In a study, the combination of *Lb. delbrueckii* subsp. *bulgaricus* (IMAU20401) and *S. thermophilus* ND03 strains (the ratio was 1:1000) has been shown to be the most optimal value for the production of aldehydes and ketones that contribute significantly to flavour [70]. Dan et al., (2019) emphasized that the flavouring substances were at the highest level when the starter ratio of *Lb. plantarum* P-8 to yogurt starter culture was 1:100. Therefore, *Lb. plantarum* P-8 strain can be used with yogurt starter culture as it does not adversely affect the physicochemical characteristics of the product [66]. The dough fermentation with *Lb. plantarum* and *Lactobacillus casei* improves soy-flour nutrient content and organic acid production together with the rheological and physicochemical properties of the dough [57].

Although rice and wheat bran are rich in fiber, protein and starch, they are expressed as the main wastes of wheat and rice processing. The odor intensity of rice and wheat bran fermented with *Lb. plantarum* 423 is increased, particularly for sulphides and aromatics [63]. The riboflavin (76–113%) and folate (32–60%) content of the cauliflower–white bean mixture increased after being fermented with *Lb. plantarum* strains (299v, Lp900, 299, Heal19). Furthermore, a remarkable (66%) rise in vitamin B_12_ was detected in *Lb. plantarum* 299 [6]. Antioxidant properties of 11 *Lb. plantarum* strain isolated from traditional Chinese fermented foods were evaluated. *Lb. plantarum* C88 (10^10^ CFU/mL) showed the highest hydroxyl radical and 2,2-diphenyl-1-picrylhydrazyl scavenging activities and thus it has been stated that it should be considered a potential antioxidant in functional foods [60]. 

## 6. Safety Aspects of *Lactiplantibacillus plantarum* including Novel Pathway for Bacteriocin Production

LAB have usually been distinguished as safe for animal and human consumption [10,18]. The use of any new microbial strain in food must guarantee its safety and toxicity under review of existing regulations [71]. Therefore, many factors should be evaluated to determine the safety of any *Lb. plantarum* strain. Among these elements, it is worth highlighting the identification of virulence factors and toxin genes, as well as the presence of mobile genetic components such as plasmids and bacteriophages in order to prevent intercellular genetic exchange with other pathogenic microorganisms [72,73]. However, the study of the production of undesirable metabolites such as biogenic amines and D-lactate acquires the special interest, since their presence in food leads to health side effects [74] and favor the metabolic acidosis suffered by patients with short-bowel syndrome or carbohydrate malabsorption [75], respectively. Moreover, the analysis of the bile salt deconjugation capacity is important because high capacities can compromise the normal digestion of lipids, alter intestinal conditions, and induce gallstones [72]. Furthermore, both the analysis of antibiotic resistance and the study of drug production by the microbial strains are crucial to limiting the appearance of new subpopulations with resistance to antibiotics [72].

Taking into account some of the factors mentioned above, until today, most of the research endorses the safety aspect of *Lb. plantarum* [7]. For instance, Todorov et al., (2017) concluded that *Lb. plantarum* ST8Sh isolated from Bulgarian salami “Shpek” may be applied in fermented food products since this strain showed a low presence of virulence genes (only 13 genes related to sex pheromones, aggregation substance, collagen adhesion, tetracycline, gentamicin, chloramphenicol, and erythromycin were detected) during its metabolism [76]. At the same time, Yang et al., (2021) found that *Lb. plantarum* IDCC 3501 produced lower concentrations of D-lactate than other lactic acid bacteria, with the consequent benefits [73]. Besides, *Lb. plantarum* IDCC 3501 displayed the absence of harmful enzymatic activity as this strain did not have α-chymotrypsin, and the presented levels of β-glucosidase were low compared with other lactic acid bacteria [73]. For their part, Syrokou et al., (2022) observed the absence of pathogenic factors in six *Lb. plantarum* subsp. *argentoratensis* strains were isolated from spontaneously fermented Greek wheat sourdoughs since the probability of the strains being a human pathogen was found to be low in a genomic and in silico analysis [77]. Moreover, they observed that the six strains analysed were not biogenic amine producers due to the absence of key genes in their genome (with the exception of cadaverine). However, the same authors detected some antibiotic resistance genes, although the aforementioned tolerance was not experimentally validated [77]. Contrary to these findings, Evanovich et al., (2019) did not identify antibiotic resistance genes on the *Lb. plantarum* genome strains are available in the GenBank sequence database. Furthermore, these authors did not observe any virulence factors [78]. Similarly, Katiku et al., (2022) classified *Lb. plantarum* Eger202111 as sensitive to specific antibiotics and Chokesajjawatee et al., (2020) demonstrated the absence of transferable antibiotic resistance genes in the genome of *Lb. plantarum* BCC9546 in an in silico analysis [72,79].

In contrast, the safety of *Lb. plantarum* strains should also be guaranteed in in vivo studies before being employed as a food additive. Thus, several studies have shown the innocuousness of various strains of this LAB. For example, Pradhan et al., (2019) observed in an oral toxicity study in mice that short-and long-term administration of a high concentration of *Lb. plantarum* MTCC 5690 (10^12^ CFU/ animal) did not disrupt any haematological or general health parameters or cause any organ-specific disorder [71]. For their part, Yang et al., (2021) found that the *Lb. plantarum* IDCC 3501 strain did not show mortality in a murine mouse model after administration of 3.4–3.6 × 10^11^ and 2.3–3.4 × 10^12^ CFU/animal, across 14 days. In addition, in this trial, the mice did not show significant changes in behaviour, skin, food consumption or bodyweight [73]. Similarly, Mukerji et al., (2016) reported that the oral administration of a combination of three *Lb. plantarum* strains (CECT 7527, 7528, and 7529) in rats (5.55 × 10^11^ and 1.85 × 10^12^ CFU/kg/day) was not associated with any adverse effects after 90 days [80]. Besides, Tsai et al., (2014) observed in an oral toxicity assay in a Wistar rat model that the administration of multiple strains of *Lb. plantarum* for 28 days (9.0 × 10^9^ and 4.5 × 10^11^ CFU/kg/day) did not modify behaviour, feed and water consumption, growth, haematology, clinical chemistry indices, organ weights, or histopathologic analysis of the rats [81].

The safety of *Lb.s plantarum* has been generally guaranteed for different strains so that its use in food would not compromise the safety of the product. Other studies have even shown that its use in fermented foods helps to improve food safety. This fact is related to the ability of *Lb. plantarum* to inhibit the growth of certain microorganisms, including pathogens [82], thus improving the shelf life of products [83]. The antimicrobial activity displayed by *Lb. plantarum* could be due both to the competition for elemental nutrients and as a product of the synthesis of active substances [84]. Therefore, the production of bacteriocins by this LAB species is of special interest, since this small peptide is a bactericide for many Gram-positive pathogens and spoilage bacteria transferred by food, including *Listeria* spp., *Pediococcus* spp., *Staphylococcus* spp., etc. [85]. Specifically, *Lb. plantarum* produces a bacteriocin generally referred to as plantaricin, which usually belongs to class I (lantibiotic) and class II (non-lantibiotic) bacteriocins [19]. However, most of the plantaricins were obtained from *Lb. plantarum* belong to class II and subgroup b, since they are non-lantibiotic, small (<10 kDa) two-peptide molecules, hydrophobic, cationic, unmodified and stable to heat [7].

Although plantaricin is a broad-spectrum antibacterial bacteriocin, its low yield may limit its future use in the food industry [86]. For this reason, new studies about the synthesis mechanisms can help to improve its obtainment, purification and food application. Currently, the obtention of bacteriocins from *Lb. plantarum* (Figure 2) generally consists of a previous incubation of the microorganism (where the bacteriocin is produced) in de Man, Rogosa, and Sharpe (MRS) broth, at 37 °C, and subsequent centrifugation of the grown culture in order to achieve the cell-free supernatant. In addition, the pH of the cell-free supernatant is usually adjusted straight away to obtain the bacteriocin after filtration. Finally, the purification and stabilization processes can be carried out on the bacteriocins obtained that favor the preservation of antimicrobial properties [86,87,88]. Nevertheless, this general scheme must be complemented with research that allows broader knowledge to be obtained for the optimization of bacteriocin production from *Lb. plantarum*, such as the influence of incubation times, the presence of certain microorganisms or substances that stimulate peptide formation, etc. Thus, Bu et al., (2021) observed that the synthesis mechanism of plantaricin Q7 was related to the ATP-binding cassette (ABC) transport system, the quorum sensing system, as well as the proteolysis system. Additionally, these authors identified that the production of plantaricin could be induced environmentally with the use of 2% NaCl and that the groS gene was a critical gene for the synthesis of this molecule [86]. Wu et al., (2021) showed that the bacteriocin obtained from *Lb. plantarum* RUB1 could be modified through its co-culture with some specific bacteria (*Enterococcus hirae* 1003 and LWS; *Limosilactobacillus fermentum* RC4; *Lb. plantarum* B6, *L. monocytogenes* ATCC 19111 and *S. aureus* ATCC 6538) or their cell-free supernatants, which increased bacteriocin activity and expression of their related genes [89]. Similarly, bacteriocin production was increased with low (100 and 500 ng/mL) and medium (1 μg/mL) concentrations of the precursor peptide PlnA since the expression of bacteriocin-related genes increased. However, this same investigation revealed that high concentrations (50 and 200 μg/mL) of the precursor peptide PlnA inhibited bacteriocin formation by *Lb. plantarum* RUB1. Furthermore, the authors also observed that bacteriocin formation is mediated by a quorum-sensing mechanism, directly influenced by autoinducing peptides or specific strains [89]. For their part, it has been identified that the synthesis of silver nanoparticles coated with Bac23 bacteriocin was a method of stabilizing the antimicrobial power of said peptide since the nanoparticles exhibited a better antimicrobial spectrum than the bacteriocin alone [88].

Consequently, the use of plantaricins obtained from *Lb. plantarum* are not currently authorized as food additives since at present nisin (Nisaplin^®^) is the only bacteriocin approved by the FDA [25,90]. However, its presence in foods can be manifested due to the direct incorporation of the producing bacteria [91]. Despite this, the safety of plantaricins should continue to be studied in depth to corroborate their safety and suitability as natural bio-preservatives in food.

## 7. New Research on Foodomics of Some Functional Fermented Foods Using *Lactiplantibacillus plantarum*

Research carried out on LAB, including *Lb. plantarum*, has usually led to a reductionist approximation working with pure culture strains, thus providing limited knowledge on understanding the impact of these bacteria on complex systems. Therefore, whole-genome sequencing of strains and shotgun metagenomics of intricate systems are powerful techniques that can be used to decipher the function and potential of probiotic microorganisms. In this way, a top-down, multiomics approximation has the capacity to solve the functional potential of an ecosystem into an image of what is being expressed, translated and produced [92]. Specifically, foodomics technologies such as metabolomic, metagenomic, and metaproteomic are now extensively employed individually or in combination and accompanied by chemometric to achieve deep insight into the role, adaptation, and exploitation of microbiota in distinct complex ecosystems, especially with regard to the production of metabolites [93].

Several studies have been conducted on fermented functional foods, with *Lb. plantarum* is being used as a culture in order to identify new compounds associated with the functional qualities of the fermented foods (Table 2). Thus, for instance, the use of Ultra-Performance Liquid Chromatography-Quadrupole Time-Of-Flight Mass Spectrometry (UPLC-Q-TOF-MS) has allowed the identification of the substances D-phenyllactic acid (PLA) and *p*-OH-PLA in green tea fermented with *Lb. plantarum* 299V [94]. The aforementioned compounds are two unique metabolites synthesized by this LAB, which have bioactive and antifungal properties. In addition, the co-cultivation of green tea with *Saccharomyces boulardii* CNCM I-745 increased the production of the two metabolites synthesized by *Lb. plantarum* 299V, which could improve the quality and preservation of fermented green teas [94].

However, the use of proteomic techniques, such as two-dimensional gel electrophoresis (2-DE) and the Matrix-Assisted Laser Desorption/ Ionization source and Tandem Time-of-Flight Mass Spectrometry (MALDI-TOF/TOF-MS) have also been employed to characterize the probiotic potential of *Lb. plantarum* S11T3E isolated from fermented olives and their brine [95]. In this way, it has been possible to confirm the probiotic properties of this strain, postulating itself as a good candidate to be described and utilized as a probiotic. This occurrence was due to the fact that in the analysis of the extracellular proteome, diverse extracellular proteins were identified (namely adherence protein with chitin-binding domain, glyceraldehyde 3-P dehydrogenase, M23 family peptidase), which are involved with adhesion processes that would be related to the ability of *Lb. plantarum* S11T3E to adhere to the gut mucosa of the host after ingestion and thus with its probiotic nature [95].

The use of foodomics techniques has also recently been employed in fermented dairy products [98]. This is the case in the research carried out by Li et al., (2021) on fermented milk, where the microbial interactions between co-cultures of *S. thermophilus* with potential probiotics, including *Lb. plantarum*, were studied under a metabolomic-based analysis [96]. Specifically, an untargeted metabolomics approach based on Ultra-High-Performance Liquid Chromatography coupled with Mass Spectrometry (UHPLC-Orbitrap MS) was utilized to map the general metabolite profiles of fermented milk. Thus, a total of 179 significant metabolites were described (containing nucleosides, amino acids, short peptides, organic acids, lipid derivatives, carbohydrates, carbonyl compounds, and substances associated with energy metabolism). The UHPLC-Orbitrap MS technique allowed the conclusion that the co-culture of *Lb. plantarum* with *S. thermophilus* showed a higher metabolic profile than the co-culture of *Bifidobacterium animalis* ssp. *lactis* together with *S. thermophilus* during the 21 days of storage at 4 ºC. In addition, the same authors concluded that the profile of the metabolites that typify the fermented milk samples depend on the starter cultures, and the inclusion of probiotic cultures such as *Lb. plantarum* considerably affects the metabolomic activities of the fermented milk [96].

However, Zha et al., (2021) evaluated the changes in *Lb. plantarum* P9 fermented milk metabolomes during its fermentation and storage, employing Ultra-Performance Liquid Chromatography-Quadrupole coupled with Time-of-Flight Mass Spectrometry (UPLC-Q-TOF-MS/MS) [97]. This analysis evidenced various changes in the milk metabolome after the fermentation process and its subsequent storage for 28 days at 4 °C. Specifically, they identified 35 metabolites, of which 25 were increased with fermentation, while 10 were decreased after the process. Among these metabolites, fatty acids, peptides, and carbohydrates were found, some of them being able to show functional characteristics in the final foodstuff. In addition, in this research, it was observed that various fatty acids, such as stearic, 3-phenyllactic, 10-ketostearic, and 10-hydroxystearic acids, as well as some bioactive molecules, were strongly affected during the fermentation and storage of *Lb. plantarum* P9 fermented milk [97]. Thus, knowledge about the influence of metabolites throughout milk fermentation and storage could improve the development of functional fermented dairy products through the use of *Lb. plantarum* P9 strain [97].

## 8. Health impacts of *Lactiplantibacillus plantarum*

It has been reported in many in vitro and in vivo studies that *Lb. plantarum* has health-promoting benefits besides its functional properties in the food industry [99,100,101]. The *Lb. plantarum* strains which have probiotic potentials may improve intestinal microbiota, regulate the immune system, reduce blood cholesterol levels and the risk of some cancers [102]. Organic acids such as phenyllactic acid, hydroxyphenyllactic acid, lactic acid, and indole lactic acid from *Lb. plantarum* UM55 may reduce the risk of cancer by inhibiting the production of aflatoxins, which are reported to have a potential relationship with cancer [103]. Yamane et al., (2018) also showed that kefir containing six different LAB, including *Lb. plantarum,* increased the cytotoxicity of human natural killer (NK) cells as well as the expression and secretion of interferon-gamma (IFN-γ) in NK cells [104]. IFN-γ has improved not only the cytotoxicity of colorectal tumour HCT116 cells but also human chronic myelogenous leukemia K562 cells [104]. In addition to *Lb. plantarum*, its extracellular polysaccharides may inhibit the proliferation of colorectal cancer cells [105].

Since obesity is becoming a global public health problem, the importance of safe and healthy non-drug treatment approaches has also increased. In this context, the use of probiotics is one of the most popular topics in recent studies [106,107]. Choi et al., (2020) showed that *Lb. plantarum* LMT1-48 had anti-obesity effects in high-fat diet-induced obese mice. This strain (at least 10^6^ CFU) downregulated the expression of lipogenic genes including *PPARγ*, *C/EBPα*, *FAS*, and *FABP4* as well as reduced the body and fat weight in obese mice [106]. Acid and bile salt tolerance, high cell adhesion activities and lipid metabolism-regulating capabilities are reported in *Lb. plantarum* KLDS1.0344 and KLDS1.0386 strains. In another study, the combination of *Lb. plantarum* KLDS1.0344 and KLDS1.0386 strains have been found to inhibit the formation of high fat-induced obesity by improving the obesity-related indicators such as body weight, body fat weight and Lee’s index [108]. Recently, the mixture of the same *Lb. plantarum* strains (KLDS1.0344 and KLDS1.0386) exhibited similar beneficial effects on obesity [107]. Unlike the previous study, the role of intestinal microbiota was investigated. These strains could manipulate the intestinal microbiota and its metabolites, which resulted in inhibition of obesity, reduction of liver lipid accumulation and improvement of lipid metabolism [107]. *Lb. plantarum* KFY02 isolated from the naturally fermented milk yoghurt could effectively treat obesity in mice fed with a high-fat diet via the *PPAR-α/γ* signalling pathway [109]. On the other hand, *Lb. plantarum* may improve the stability of the intestinal tract and suppress the proinflammatory cytokines during the development of inflammatory bowel diseases [110]. *Lb. plantarum* 299v may support the treatment of cancer, irritable bowel syndrome, and *Clostridium difficile* infection as well as it may make positive alterations in the composition of the human gut microbiome and the immune system [111]. Given the health-promoting effects reported so far, *Lb. plantarum* strains deserve further studies related to their potential health benefits and risks.

## 9. Conclusions

*Lb. plantarum*, besides being a well-characterized probiotic bacterium, is a versatile microorganism with the ability to improve the functional properties of fermented foods, offering various applications in the food industry. *Lb. plantarum* is stably found in various niches, however, its usual habitat is dairy products, plants, fruit flies, and the digestive tract of vertebrates. *Lb. plantarum* strains have the ability to improve the nutritional quality, antioxidant activity and flavor properties of foods along with antimicrobial activities, reducing undesirable compounds and improving the shelf life. Since *Lb. plantarum* is listed as GRAS, its bacteriocins are also considered safe to use in the food industry as bio-preservatives. Moreover, according to the genome sequence of *Lb. plantarum*, no pathogenic or antibiotic resistance genes were identified in *Lb. plantarum*. Still, more whole-genome sequencing of strains and shotgun metagenomics studies are required to understand the function and potential use of *Lb. plantarum* strains. 

## Figures and Tables

**Figure 1 microorganisms-10-00826-f001:**
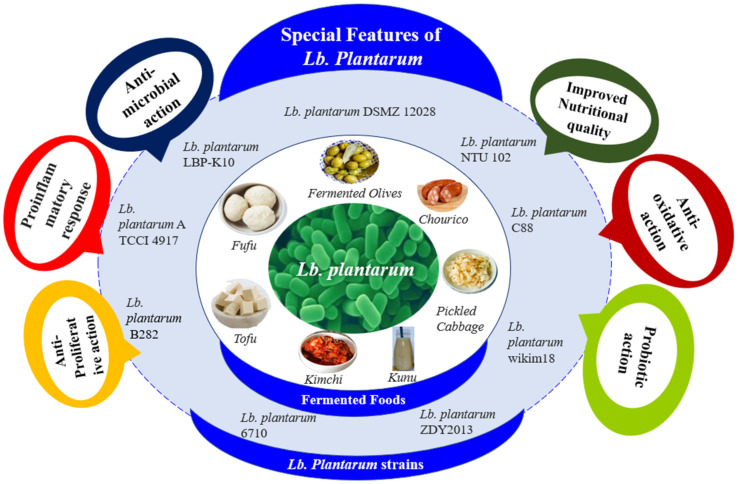
The functionality of *Lb. plantarum* strains.

**Figure 2 microorganisms-10-00826-f002:**
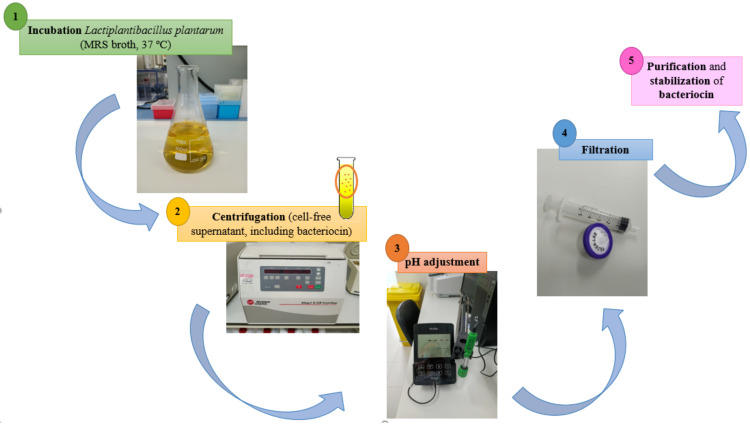
General scheme of bacteriocin production from *Lactiplantibacillus plantarum.*

**Table 1 microorganisms-10-00826-t001:** Functional properties of *Lactiplantibacillus plantarum* in fermented foods.

Fermented Foods	*Lb. plantarum* Strain(s)	Application in Food Industry	Functional Impacts	Reference
Rice and wheat bran	*Lb. plantarum* 423	Antioxidant activity and flavour properties	-Fermentation improved the hydroxyl radical-scavenging activity and oxygen radical-scavenging activity. -It also enhanced odor intensity.	[63]
Wheat fermented silage	*Lb. plantarum* QZ227	Fermentation profile and microbiological composition	-*Lb. plantarum* QZ227 showed good probiotics features (good stress tolerance of temperature, bile, salt, acid, and alkali). -It could efficiently suppress various pathogens found in silage.	[64]
Cauliflower and white beans	*Lb. plantarum* 299v, Lp900, 299, Heal19	Improving the vitamins and amino acid composition	-When compared to an unfermented control, all strains considerably enhanced folate and riboflavin levels. -*Lb. plantarum* 299 significantly increased the vitamin B_12_ content while it improved amino acid content slightly.	[6]
Fresh-cut apples	*Lb. plantarum* BX62 (alone or in combination with chitosan)	Improving the qualitative characteristics as a bio-preservative	*Lb. plantarum* BX62 (in combination with chitosan), significantly reduced the counts of aerobic mesophilic bacteria, aerobic psychrophilic bacteria, yeast, and molds.	[65]
Fermented milk	*Lb. plantarum* P-8	Fermented milk flavour and storage stability	The 1:100 ratio of *Lb. plantarum* P-8 to yogurt starter cultures improved the stability and volatile flavour compounds of fermented milk.	[66]
Yogurt	9 *Lb. plantarum* strains	Fermentation properties and subsequent changes	*-Lb. plantarum* IMAU80106, IMAU10216, and IMAU70095 showed the highest coagulation ability and proteolytic activity. -*Lb. plantarum* IMAU70095 had the best results in terms of the texture and volatile flavour profiles.	[67]
Kimchi	*Lb. plantarum* PL62	Food quality and microbiota of Chinese cabbages kimchi	-*Lb. plantarum* PL62 was found on the first day of fermentation and during the entire 25-day fermentation. -The survival of *Lb. plantarum* PL62 during fermentation suggests that a functional probiotic might be introduced to a variety of fermented foods.	[68]
Traditional Chinese fermented dairy tofu	11 *Lb. plantarum* strains	Antioxidant activity	*Lb. plantarum* C88 showed the highest hydroxyl radical and DPPH scavenging activities as well as it was the most resistant strain against hydrogen peroxide.	[60]
Traditional Chinese fermented dairy tofu	*Lb. plantarum* C88	Reduction of aflatoxin B1 toxicity	The strongest aflatoxin B1 binding capacity was found in *Lb. plantarum* C88 as well as it increased antioxidant capacity.	[61]
Spontaneously fermented carrots	*Lb. plantarum* 299v	Food safety and quality	*Lb. plantarum* 299v suppressed *Salmonella* contamination and *Enterobacteriaceae* levels.	[69]

**Table 2 microorganisms-10-00826-t002:** Application of omics technologies in the study of some functional fermented foods using *Lb. plantarum.*

Fermented Food	*Lb. plantarum* Strain	Omic Technology	Metabolites Identified	Functional Properties	Reference
Green tea	*Lb. plantarum* 299V	UPLC-Q-TOF-MS	D-phenyllactic acid (PLA) and p-OH-D-phenyllactic acid (exclusive to this strain)	Bioactive and antifungal properties	[94]
Olives and olives brine	*Lb. plantarum* S11T3E	2-DE and MALDI-TOF/TOF-MS	Extracellular proteins involved in adhesion processes	Ensures adhesion to the host mucosa	[95]
Fermented milk	*Lb. plantarum*	UHPLC-Orbitrap MS	Identification of 179 different metabolites	The large abundance of beneficial metabolites	[96]
Fermented milk	*Lb. plantarum* P9	UPLC-Q-TOF-MS/MS	Identification of 35 different metabolites (including fatty acids, peptides, and carbohydrates)	Metabolites with functional properties	[97]

## Data Availability

Not applicable.

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
