# Peer review of "The Impacts of Lactiplantibacillus plantarum on the Functional Properties of Fermented Foods: A Review of Current Knowledge"

_microorganisms, 2022, doi:10.3390/microorganisms10040826_

Round 1

Reviewer 1 Report

The present review paper presents an interesting and comprehensive study regarding Lactiplantibacillus plantarum, its genomic mapping, gene locations, and utilisations, especially in functional foods.

The study is good, but it has to be improved, as it is quite confusing in some places. How was the study conducted, and how were the articles selected? Which were the basic criteria?

Otherwise:

  • in abstract line 29 please correct dairy to dairy products,
  • from line 96 - this section is too focused on some studies (96 - 110 - ref 24; 111-122 ref 25, and line 123 - 133 ref 13), maybe their correlation with other studies would be helpful. i.e. 10.3389/fmicb.2018.00893; 10.1016/j.lwt.2021.113054; 10.1016/j.fm.2022.103989
  • line 79 "qualified presumption of safety" - This was already abbreviated at line 59
  • line 82 - lactic acid bacteria was already abbreviated above, please if the authors abbreviate something then use it afterward
  • I think this section should cite different references, as this reference was also before introduced in line 237. 
    i.e. the authors could cite a recent study that analyzed the effect of Lb. plantarum in dough fermentation: https://doi.org/10.3390/foods9121894 
  • table 1 - how were the studies selected? There are many more studies with several strains of Lb. plantarum, why were they excluded, or why were these included?
  • line 281 - 282 - This aspect has already been mentioned in the introduction (lines 57 - 62). Please don't mention the same aspect several times. Revise the whole manuscript.
  • line 407 - 409 - Please rephrase
  • please correct the references, and use italics

Based on these aspects, I suggest major revisions.

Author Response

Dear Reviewer,

Firstly, thank you so much for your valuable comments and explanations and your great suggestions and for giving us a chance to improve our manuscript. We have tried to edit our manuscript according to your revision. The red colour font has been used to indicate the corrections or modifications made in the manuscript. Also, you can find the corrections one by one below. We hope our manuscript is improved after your critical revision.

Please see our corrections in the attachment.

Best regards,

Reviewer 2 Report

Dear Editor,

Thank you for the opportunity to review the paper entitled "The Impacts of Lactiplantibacillus plantarum on the Functional Properties of Fermented Food: A Review of Current Knowledge".  I find this review very interesting and deeply informative from the scientific point of view. Nevertheless, I have some suggestions for the paper quality improvement:

  • The paper is well-structured, but in my opinion it is necessary to add a special section about Lactiplantibacillus plantarum health impact. Below, you may find some references related to this subject.

Mariadhas Valan Arasu, NaifAbdullah Al-Dhabi, Soundharrajan Ilavenil, Ki Choon Choi, Srisesharam Srigopalram, In vitro importance of probiotic Lactobacillus plantarum related to medical field, Saudi. J. Biol. Sci. (2016) 23 (1) S6-S10.

Karolina Kaźmierczak-Siedlecka, Agnieszka Daca, Marcin Folwarski, Jacek M. Witkowski, Ewa Bryl, Wojciech Makarewicz, The role of Lactobacillus plantarum 299v in supporting treatment of selected diseases, Cent. Eur. J. Immunol. (2020) 45 (4) 488-493.

Amro Abdelazez, Heba Abdelmotaal, Zong-Tao Zhu, Jia Fang-Fang, Rokayya Sami, Lu-ji Zhang, Abdel Rahman Al-Tawaha, Xiang-Chen Meng, Potential benefits of Lactobacillus plantarum as probiotic and its advantages in human health and industrial applications: A review, Adv. Environ. Biol. (2018) 12 (1) 16-27.

  • I would like to point out the good choice, diversity and quantity of analysed literature
  • The figures in the manuscript should be presented in better quality form.

Author Response

Dear Reviewer,

Firstly, thank you so much for your valuable comments and explanations and your great suggestions and for giving us a chance to improve our manuscript. We have tried to edit our manuscript according to your revision. The blue colour font has been used to indicate the corrections or modifications made in the manuscript. Also, you can find the corrections one by one below. We hope our manuscript is improved after your critical revision.

Please see our corrections in the attachment.

Best regards,

Round 2

Reviewer 1 Report

The article was very well improved, and every suggested recommendation was introduced. Based on these aspects I consider that the article can be published.